# Long-Range Thermal Target Detection in Data-Limited Settings Using Restricted Receptive Fields

**DOI:** 10.3390/s23187806

**Published:** 2023-09-11

**Authors:** Domenick Poster, Shuowen Hu, Nasser M. Nasrabadi

**Affiliations:** 1Lane Department of Computer Science and Electrical Engineering, West Virginia University, 395 Evansdale Dr., Morgantown, WV 26506, USA; nasser.nasrabadi@mail.wvu.edu; 2DEVCOM Army Research Laboratory, 2800 Powder Mill Rd., Adelphi, MD 20783, USA

**Keywords:** deep learning, small object detection, automated target recognition, thermal infrared

## Abstract

Long-range target detection in thermal infrared imagery is a challenging research problem due to the low resolution and limited detail captured by thermal sensors. The limited size and variability in thermal image datasets for small target detection is also a major constraint for the development of accurate and robust detection algorithms. To address both the sensor and data constraints, we propose a novel convolutional neural network (CNN) feature extraction architecture designed for small object detection in data-limited settings. More specifically, we focus on long-range ground-based thermal vehicle detection, but also show the effectiveness of the proposed algorithm on drone and satellite aerial imagery. The design of the proposed architecture is inspired by an analysis of popular object detectors as well as custom-designed networks. We find that restricted receptive fields (rather than more globalized features, as is the trend), along with less downsampling of feature maps and attenuated processing of fine-grained features, lead to greatly improved detection rates while mitigating the model’s capacity to overfit on small or poorly varied datasets. Our approach achieves state-of-the-art results on the Defense Systems Information Analysis Center (DSIAC) automated target recognition (ATR) and the Tiny Object Detection in Aerial Images (AI-TOD) datasets.

## 1. Introduction

Thermal object detection, or the detection of objects in thermal imagery, is an attractive research area because of the unique advantages offered by the use of thermal cameras. Thermal cameras are able to sense heat signatures in total darkness and, with the appropriate long-range sensor, from distances of 5 km away or more. Thermal sensors are useful for stealth reconnaissance as they measure thermal infrared radiation in a completely passive fashion. Furthermore, they can also be employed to complement scene information obtained by other devices, such as visible spectrum or Lidar sensors. Given the utility of thermal sensors, thermal object detection can be deployed in a wide range of civilian, industrial, and scientific applications, such as automated driving and wildlife monitoring, or in military and defense scenarios, such as surveillance and battlefield target detection. The research presented herein primarily explores the latter scenario of battlefield target detection. However, the overall aim is to develop an approach generally suitable for single-frame small object detection in data-limited settings.

Long-range thermal object detection is a challenging research problem because it adds further complications to the nontrivial task of generic object detection. These additional complications can be summarized into two fundamental challenges:The *signal challenge*: Thermal sensors are limited by the resolution and level of detail they can capture. The detection of very small or distant objects exacerbates the challenges inherent to the thermal domain—namely the object’s very low resolution, or “pixels on target” (PoT).The *data challenge*: The current state of the art in object detection involves highly data-driven neural network algorithms. However, the reality is that specific applications involving long-range thermal object detection likely do not have the luxury of abundant labelled training data matching the distribution of data on which the trained model is expected to realistically operate. Thermal background and target signatures also exhibit non-homogenous properties due to variable atmospheric conditions, such as daylight, nighttime, or cloudy skies.

The way in which these issues manifest in modern neural network object detection models is two-fold. First, the common backbone convolutional neural network (CNN) architectures, such as DarkNet [1,2,3], Inception [4,5,6], ResNet [7], ResNext [8], and MobileNet [9], perform early spatial downsampling of feature maps. In the case of very small objects with low PoT, early downsampling can cause premature loss of information, destroying the already limited signal that exists. Secondly, state-of-the-art (SOTA) object detection models have a large number of parameters and very large receptive fields, often easily covering an area of 1024×1024 pixels or more. While this allows for the representation of semantically rich features with large spatial context, it also increases the model’s capacity for overfitting in data-limited scenarios.

In order to address the issues arising from the signal and data limitations for long-range thermal object detection, we propose a novel backbone architecture customized for the detection of distant or small objects in thermal imagery. By carefully managing the interplay between the network’s receptive field and feature map resolution, the proposed model achieves better generalizability in unseen target contexts and higher detection accuracy compared to SOTA algorithms on long-range thermal target detection.

The proposed model is evaluated alongside several baseline object detection algorithms on the largest public thermal image dataset of civilian and military vehicles currently available—the Defense Systems Information Analysis Center (DSIAC) Automated Target Recognition (ATR) dataset [10]. To gauge model generalizability and robustness, we define a challenging training and testing protocol for the DSIAC dataset. We also present the results of a series of architectural ablation studies that validate our design methodology.

To summarize, the contributions of this paper are

Proposal and validation of a novel architecture designed for very long-range/small thermal target detection, especially in data-limited settings.Design of a challenging train/test protocol for the DSIAC dataset.Performance analysis of our proposed model alongside SOTA object detection models on the DSIAC and Tiny Object Detection in Aerial Images (AI-TOD) datasets.

The remainder of the paper is organized as follows. Section 2 discusses prior works in ATR and CNN-based object detection. In Section 3, we describe the DSIAC dataset and our proposed training and evaluation protocol. Next, Section 4 evaluates common SOTA object detector baseline models on the DSIAC dataset and explores the issue of background overfitting. Section 5 presents our proposed architecture with several ablation studies validating the design methodology. An overall comparison of our model with existing SOTA methods is provided in Section 6. Finally, Section 7 summarizes our findings and briefly discusses future work.

## 2. Related Work

### 2.1. ATR Pre-Deep Learning

Due to the limited amount of training data for automatic target recognition research, especially in the thermal infrared domain, as well as a lack of advancement in neural network algorithms (the two being interrelated), most prior automated target recognition (ATR) work relies on hand-crafted feature extraction and/or statistical image preprocessing methods in order to discern and classify target signals from background regions.

Gregoris et al. [11] and Der et al. [12] use double-gated windows to compare the structural information of an inner window to an outer window in order to detect possible targets. In [11], the ratio of the mean difference of the inner and outer windows’ pixel value to the standard deviation of the outer window is compared to a threshold value (threshold values and window sizes are preassigned hyperparameters). A combination of features, such as contrast difference, magnitude of gradients, and edge information, are used in [12] to compare inner and outer windows. Yoon et al. [13] preprocess input images using localized pixel thresholding followed by a Sobel edge detector and then train a Bayesian classifier on the normalized moment of inertia calculated from image patches. Mahalanobis et al. [14] propose a quadratic correlation filter—a set of basic functions estimated from the second-order statistics—to localize and classify targets.

Zhou and Crawshaw [15] extract low- and high-resolution Gabor features to first filter out background clutter and then refine the binary target prediction. Targets are identified by comparing the extracted features to the Gabor features of a set of five basic geometric patterns (rectangle, square, oval, rounded-rectangle, and circle). Histogram of oriented gradients (HOG) features and a support vector machine classifier are proposed in [16]. Gray et al. [17] match scale-invariant feature transform (SIFT) features from input images to a target gallery using a Euclidean distance measure.

In order to address the problem of data and target non-homogeneity, Zhang et al. [18] propose a feature-agnostic dictionary learning method using a joint sparse representation constraint.

### 2.2. Deep-Learning-Based Object Detection

Current state-of-the-art object detection models are based on neural network architectures that learn to perform the entire detection process in a more integrated fashion. Feature extraction, target localization, classification, and even aspects of pre-processing, such as image normalization, are all learned in an end-to-end manner. The model is primarily composed of a series of parameterized, learnable, nonlinear transformations. As a result, these models are able to learn rich high-level semantic representations. However, the highly parameterized nature of these networks creates a need for abundant training samples that well represent the data distribution on which they are expected to perform. Convolutional neural networks (CNN) typically form the backbone of modern object detection models. Recently, large-scale vision transformer models [19,20,21] have yielded impressive results but require training on massive datasets.

Object detection algorithms are typically grouped into two binary categories: (1) one-stage or two-stage, and (2) anchor-based or anchor-free. Two-stage (Faster R-CNN [22]) detectors first propose regions of interest (RoI) via an initial binary classification (object vs. background) and bounding box regression phase. Cropped and resized sections of feature maps corresponding to proposed RoIs are input into a secondary set of layers specialized for multi-class prediction and bounding box refinement. One-stage detectors (YOLO [1], single-shot detector (SSD) [23], and RetinaNet [24]) forego the secondary set of layers in favor of directly performing multi-class classification on the extracted features.

Anchor-based models (such as the aforementioned one-stage detectors) regress bounding box coordinates as offset values in relation to sets of rectangular-shape priors known as anchor boxes. Anchor boxes are hyperparameters that must be carefully chosen based on the expected range of target sizes. For medium- to long-range ATR, where the size and shape of targets tend to vary less than in general object detection applications, the choice of anchor boxes may be less daunting. Anchor-free is a catch-all term for models that use alternative methods for bounding box regression, such as keypoint detection. Examples include CornerNet [25], CenterNet [26], and fully convolutional one-stage detector (FCOS) [27].

### 2.3. Deep-Learning-Based ATR

There is little existing thermal ATR research using modern neural network architectures. DeepTarget [28] is a two-stage ATR model composed of two separate fully convolutional VGG-like [29] convolutional neural networks (CNN). The first CNN is trained as a binary classifier on target and background image patches, functioning as an initial background clutter rejection stage. Designed as a fully convolutional network (lacking any fixed fully connected layers), it produces a heatmap of possible target locations when given a full image. The potential target locations become input patches for the second CNN, which is trained as a multi-class classifier. In this way, each CNN specializes in either clutter rejection or fine-grain classification. However, DeepTarget’s two separate stages are not jointly optimized, as occurs with Faster R-CNN and most other two-stage detectors. Nevertheless, the reported results, benchmarked on the Comanche (BoeingSikorsky, USA) ATR FLIR dataset, show DeepTarget outperforming contemporary state-of-the-art object detectors such as Faster R-CNN, YOLOv2, and SSD, as well as other non-neural-net state-of-the-art ATR algorithms. Unfortunately, the Comanche dataset does not contain long-range targets, nor was it available for public use at the time of our work.

Mahalanobis and McIntosh [30] compare Faster R-CNN [22] to QCF [14] on the DSIAC [10] dataset (see Figure 1 for sample images and Section 3 for more details on DSIAC). An “easy” and “difficult” test set are designed, both composed of 19,800 images randomly sampled across all ranges. The “difficult” set only contains targets in the lowest quartile of both target size and contrast regarding the nearby background. A disjoint training set is constructed by uniformly sampling every 100 frames across all ranges. Faster R-CNN is reported to significantly outperform QCF, achieving an 80% probability of detection (PD) at a false alarm rate (FAR) of 0.01 false alarms per square degree on the difficult set compared to QCF’s 45% PD at FAR = 0.4 false alarms/sq degree.

Chen et al. [31] benchmark three YOLOv2-based architectures [2] on the DSIAC dataset: a deeper 74-layer network using a ResNet-50 backbone [7] and two shallower 22- and 25-layer networks with custom CNN backbones. Further, 1620 frames are evenly sampled for each of four different targets at each range to create a pool of images used for training and testing. The authors also manually inspected the bounding box annotations in order to correct errors. Results are reported on several experimental scenarios, including day vs. night performance, training on a variable number of images, ImageNet pre-training, and performance on different sets of ranges. The authors report better performance on nighttime scenarios due to the greater contrast of target and background heat signatures. The shallower non-pretrained CNNs also achieve slightly better performance than the deeper pre-trained ResNet backbone. However, due to the high correlation between images of the same video sequence, all three architectures achieve very high performance (more than 96% mean average precision) even at the farthest ranges, thus prompting the need for a more challenging evaluation protocol.

The use of synthetically generated thermal target signatures for training an ATR system is explored by d’Acremont et al. [32]. They utilize the OKTAL-SE software, which can simulate novel thermal scenes based on real target signature image sequences and atmospheric metadata. The real training and evaluation data were taken from the DSIAC dataset, but the range scenarios that were used are not specified. Their custom-designed “cfCNN” model contains seven convolutional layers with 5 × 5 kernels. Max pooling occurs after the second, fourth, and fifth layers. Similar to DeepTarget [28], cfCNN is a patch-based classifier trained on 128 × 128 background and target crops, outputting a heatmap of class probabilities when evaluated on full images. As such, it has similar disadvantages as DeepTarget in that the coarse patch-based predictions can struggle to precisely localize small targets or differentiate clustered targets.

In support of the thermal ATR task, Abraham et al. [33] improve the performance of YOLOv5 models on the DSIAC dataset via a novel homotopy-based hyperparameter optimization algorithm. Leveraging the visible imagery of the DSIAC dataset, VS et al. [34] propose a metalearning strategy for unsupervised domain adaptation in thermal ATR.

Range estimation in thermal imagery is an open problem that could assist scale-aware long-distance thermal ATR. Bao et al. [35] introduce heat-assisted detection and ranging (HADAR) to perceive texture, depth, and decluttered physical attributes from thermal infrared images. HADAR is developed and tested primarily for closer-range automated driving tasks as opposed to long-range scenarios but still represents a promising thread for thermal detection research.

### 2.4. Small and Tiny Object Detection

Long-distance thermal ATR is closely related to other work in small and tiny object detection.

**Datasets:** Applicable datasets predominantly involve aerial imagery from drones or satellites. The BIRDSAI [36] dataset contains aerial imagery (62,000 real and 100,000 synthetic frames) from a drone-mounted thermal camera monitoring humans and animals in a nature reserve. Approximately 10–20% of real and 50% of synthetic images contain target bounding boxes with areas less than 200 pixels, the vast majority of which belong to the human class. None of the data are taken at distances on the scale of the DSIAC dataset. Rather, a large percentage of the data contain bounding box sizes larger than 2000 pixels, making the dataset better suited to multi-scale approaches.

The VisDrone [37] dataset contains 272,117 visible-wavelength drone photographs of pedestrians and vehicles in a variety of urban and rural settings. Like the BIRDSAI dataset, the vast majority of objects are closer than 1 km from the camera and a large number of bounding boxes are on the order of 1000 pixels in size or larger.

The AI-TOD [38] is a curated aggregation of drone and satellite images from the VisDrone2018, DOTAv1.5, xView, Airbus-Ship, and DIOR datasets. Containing twenty eight thousand thirty six aerial images, eight classes, and over seven hundred thousand object instances with a mean bounding box size of 12.8 pixels, the AI-TOD dataset focuses on tiny object detection. Given the composite nature of the dataset, objects and backgrounds are highly diverse in appearance—a luxury not currently available for long-range thermal ATR research.

**Approaches:** Several types of strategies have been proposed to deal with the detection of small and tiny objects. These can be broadly categorized into label assignment strategies, metrics and loss functions, and feature enhancement methods.

Imbalanced proportions of foreground and background target instances is an issue for object detection in general that can worsen when the majority of images contain large background regions and sparse distributions of small objects. Zhang et al. [39] propose an adaptive label assignment strategy based on ground truth and anchor box intersection over union (IoU) statistics to achieve a better foreground/background balance. Another approach proposed by Kim et al. [40] is to dynamically learn a Gaussian distribution of positive and negative anchor boxes as they relate to their impact on classification and localization losses.

The IoU metric underlies the label assignment, localization loss, and non-max suppression (NMS) components of object detection. Bounding box predictions have an increased change to have zero overlap with the ground truth when targets are tiny. Generalized IoU (GIoU) attempts to smooth the learning of bounding box predictions by modifying the IoU metric such that two non-overlapping boxes yield a non-zero metric based on the size of the smallest mutually overlapping box. Furthermore, IoU is more sensitive at small scales. Therefore, the normalized Wasserstein distance [41], which has the additional benefit of scale-invariance, has been proposed as a replacement. Lin et al. [24] address the class imbalance issue via a focal loss term that prioritizes the learning of hard samples.

Super-resolution is a particularly attractive strategy for small object detection. Several generative adversarial network (GAN) methods have been proposed [42,43,44,45]. Recent works have also investigated super-resolution of thermal images for the purpose of object detection and segmentation [46,47]. To our knowledge, no studies have attempted to specifically super-resolve and detect tiny thermal signatures. Regardless, given current data and technological constraints, the amount of recoverable detail from a 15-pixel thermal target is likely to be very limited.

Improved utilization and enhancement of fine-grained details can also be facilitated via multi-scale architectural components that create multiple information pathways to merge low- and high-resolution features, such as feature pyramid network (FPN) [48], path aggregation net (PANet) [49], and TridentNet [50]. Taking inspiration from both multi-scale and super-resolution strategies, Deng et al. [51] propose an FPN with an extended super-resolved feature layer.

A custom-designed backbone specialized for tiny object detection in aerial images proposed by Pang et al. is TinyNet [52]. It is similar to our work in that it focuses on the design of the backbone feature extraction architecture. However, TinyNet differs from our proposed approach in several key aspects. First, it has significantly fewer kernels per layer (ranging from 18 to 72), resulting in highly compressed feature representations. Second, it retains the aggressive downsampling schedule of other popular backbones, reducing a 640 × 640 image to a 40 × 40 feature map at the deepest layer. Conversely, our proposed architecture has a more traditional progression of channel depth (from 32 to 256 kernels), less downsampling of feature maps, narrower receptive field, and shorter network stride.

## 3. DSIAC Dataset

The DSIAC ATR Algorithm Development Image Database [10] is a large collection of visible and mid-wave infrared (MWIR) imagery collected by the US Army Night Vision and Electronic Sensors Directorate (NVESD). The images represent signature data (unobstructed views taken in favorable conditions) of ten ground-based vehicles, including foreign military vehicles, one towed artillery piece, as well as dismounted civilians. The image frames are from a set of video sequences recording each target individually traveling around a preset 100 m diameter circular path at distances ranging from 1000 to 5000 m in 500 m intervals. The images are captured by two forward-looking ground-based sensors: an L3 Cincinnati Electronics Night Conqueror MWIR camera using a 640×480 pixel Indium Antimonide focal plane array with a 28 micron pitch, and an Illunis visible light camera. Images were collected from both cameras during daytime and from the MWIR camera at night. For each target/range/time scenario, images were captured at 30 Hz for one minute, yielding 1680 images per sequence. The cameras were located on slightly elevated positions overlooking a valley of dry scrubland. Extensive metadata are provided, including bounding box annotations obtained from a semi-automated process involving a subset of manually annotated frames informing an automated tracking algorithm. A sample of the MWIR imagery is provided in Figure 1.

The DSIAC dataset is an invaluable resource for ATR algorithm development. However, the dataset has two important issues that we have attempted to address:The primary issue is the dataset is well-suited as a training dataset but lacks a separate test set containing the same targets but in different conditions. When training and testing on images from the same video sequence, due to the high correlation between the image frames, the detection task is trivially solved by state-of-the-art algorithms. This is demonstrated by prior works [30,31] that achieve near perfect performance at the farthest ranges despite the targets being barely perceptible. Therefore, we propose a training and testing protocol that evaluates a model’s capacity to learn the target signature without the benefit of impractically high correlation between the train and test images.The automated bounding box annotations are sometimes inaccurate at far ranges. At times, the automated tracker wanders off target before being reset by a manually annotated frame. Instead of using the provided annotations, we calculate corrected bounding boxes using positional metadata available for the MWIR imagery.

### 3.1. Proposed Training and Testing Protocol

Given the difficulty of collecting a large ATR dataset under a wide range of conditions, we argue that the ability of an ATR algorithm to handle unseen data is an important evaluation criterion. We propose a more challenging evaluation protocol for the DSIAC dataset that evaluates a model’s robustness to unseen range scenarios. The following process is used to create the proposed protocol:A subset of images is selected by subsampling every tenth frame of each video.The selected frames are randomly partitioned into ‘dev’ and ‘reserve’ sets with an 80/20% split stratified by target class, range, and time of day. The ‘reserve’ set is not used in this work and is reserved for other purposes or challenges.The dev set is subdivided into ‘train’ and ‘evaluation’ sets via 5-fold cross-validation.Finally, certain ‘testing’ ranges are designated exclusively for final model evaluation, while a disjoint set of ranges are used for both training and validation.

Training and testing on different range scenarios allow for the evaluation of model robustness to unseen background contexts. The ranges are categorized as follows:Training and Validation Ranges: {1000,2000,3000,4000,5000} mTesting Ranges: {1500,2500,3500,4500} m

Evaluation results are reported as the average performance over five models, each separately trained and evaluated on its respective fold. Additionally, we only use data from the following classes: Pickup, SUV, BTR70, BRDM2, BMP2, T72, ZSU23-4, and 2 S3. Table 1 lists the average size of the training and evaluation sets for each range and cross-validation fold. The filenames specifying our training and evaluation sets, as well as the adjusted ground truth bounding box values of the targets, are provided in the Appendix A.

### 3.2. Preprocessing

The DSIAC MWIR imagery is in 16-bit format; however, the pixel intensity values occupy a much smaller band within the full 16-bit range. The images are converted into 8-bit format via min–max normalization, stretching the range of each image’s pixel values to 0–255 and rounding to the nearest integer. Finally, the single image channel is replicated to produce a 3-channel image. No additional preprocessing is conducted to correct for any outlying values.

## 4. Analysis of One- and Two-Stage Baseline Models

In this section, we present an initial evaluation of several popular generic CNN-based object detection models: YOLOv3, YOLOv5, Faster R-CNN, and RetinaNet. These models serve as good baselines for comparison as they are simple, reliable frameworks that are competitive with the SOTA while also being excellent starting points for more customized, domain-specific solutions. The basic topology of the YOLOv3 backbone is a series of five convolutional bottleneck modules, each proceeded by a downsampling layer. The bottleneck modules encourage sparse representations by contracting and then expanding the number of filters in a set of layers. The YOLOv5 family of models follows a similar structure to YOLOv3 but replaces the standard convolutional bottleneck modules with cross-stage partial (CSP) modules [53] (discussed further in Section 5.4). The small, medium, and large variants of YOLOv5 simply vary the number of layers and filters in the CSP blocks. Both YOLOv3 and YOLOv3 employ an FPN to concatenate the upscaled lower-resolution features with the higher-resolution output heads. The two Faster R-CNN (FRCNN) and RetinaNet variants are built on ResNet backbones of either 50 or 101 layers of repeating residual convolutional blocks. All of the baseline models have three output heads corresponding to three different feature resolution scales. Analysis of these algorithms’ performance indicates two common points of failure: (1) the inability to detect very small targets (discussed in Section 4.3) and (2) an over-reliance on irrelevant background cues (discussed in Section 4.4). We argue that these failures, albeit specific to our particular long-range thermal ATR scenario, stem from a fundamental drawback of their architectural design.

### 4.1. Performance Metric

We use the mean average precision (mAP) as our performance metric, opting for the commonly used, albeit more lenient, intersection over union (IoU) threshold of 0.5 (mAP${}^{\mathrm{IoU}=0.5}$) in order to de-emphasize the importance of precise bounding box predictions on very small targets. Given a set of target classes *K*, the mean average precision at a given IoU threshold is
(1)$$\mathrm{mAP}=\frac{1}{\left|K\right|}\sum_{k\in K}{\mathrm{AP}}_{k}$$
where $AP_{k}$ is the average precision for the *k*th class. We follow the 101-point interpolated AP definition used by COCO [54].

### 4.2. Implementation Details

The models have all been trained using four Nvidia Tesla V100 GPUs. For YOLOv3 and YOLOv5, we utilize the Ultralytics YOLOv5 framework [55]. For RetinaNet and Faster RCNN models, we employ Detectron2 [56]. The pyCOCOtools framework [57] is used to calculate mAP. Unless otherwise noted, the default hyperparameter settings of the respective repositories are used. The batch size is 64 for YOLO models and 32 for Faster R-CNN and RetinaNet. The models are trained on the images from the training sets of all training ranges (1, 2, 3, 4, and 5 km). Faster R-CNN, RetinaNet, all YOLO models are trained for 300 epochs except YOLOv5 l and YOLOv5 x, which are trained for 500 epochs. The (h,w) anchor box sizes for the respective prediction heads are

(10,13), (16,30), (33,23)(30,61), (62,45), (60,120)(116,90), (156,198), (373,326).

### 4.3. Performance Analysis

The mAP (defined by Equation (Equation 1)) of the baseline models on the set of designated testing ranges is provided in Table 2. Table 2 also includes results on the validation data, where training and evaluation are conducted on the same set of ranges (in other words, on the same video sequences). As certain range scenarios are particularly challenging, model performance is subdivided by range rather than averaged together. The results on the validation ranges reflect the near-perfect detection rates reported in [30,31]. Performance on the test ranges, however, provides a truer measure of model accuracy and robustness.

Detection rates decrease dramatically on all unseen ranges except for the closest scenario. The difference between the detection rates on the validation ranges versus on the test ranges is evidence of overfitting. At 1.5 km, mAP is near 100% for several models. At 4.5 km, on the other hand, the best performing models only achieve around 20% mAP. YOLOv5 s and Faster R-CNN-50 perform best overall, while YOLOv5 x performs the worst.

Curiously, the detection rates for the 2.5 km scenario are significantly lower than 3.5 km; smaller, more distant targets *should* be more difficult to detect than larger, closer ones, all else equal. The performance decrease on the closer 2.5 km range warrants examination.

### 4.4. Background Overfitting

As evidenced in Table 2, performance decreases dramatically on unseen range scenarios. Given the limitations of the dataset and the difficulty presented by our proposed protocol, some degree of performance loss due to overfitting is to be expected. However, it is curious that detection rates for the 2.5 km scenario are significantly lower than at 3.5 km.

On close inspection, there is an important difference between the 2.5 and 3.5 km ranges that is visible upon inspection of the images. Figure 2 provides a side-by-side comparison of the range scenarios from 2 to 4.5 km meters. The views of the valley in the 3 to 4.5 km scenarios are identical (although the targets appear in different locations), whereas, in the 2 and 2.5 km scenarios, they are unique. The results suggest that the algorithms leverage familiarity with a scene to isolate the target signals (training on 3 and 4 km scenarios helps accuracy on 3.5 km). In so doing, the models become biased to particular backgrounds (training on 2 and 3 km scenarios does not generalize to 2.5 km). This could also explain why Faster R-CNN performed best in the 2.5 km scenario. The two-stage Faster R-CNN model makes final predictions based on cropped regions of interest, which may be more robust to background overfitting.

To further examine the issue of background overfitting, we copy-paste a successfully detected target along with a varying amount of its surrounding background into other areas of the image: a close crop around the vehicle’s chassis, a 50 × 50 window, and a 100 × 100 window centered around the target. Inference is run on the modified image in order to determine the “redetection rate” (i.e., can the model detect the copies?). Figure 3 illustrates the results of this experiment. The YOLOv5 model fails to redetect the majority of the cloned targets until a very large 100×100 region around the original target is also copied. One could reasonably expect the model to be able to redetect the target when it is translated into other sensible areas of the terrain. The high rate of failed redetections of YOLOv5 s is evidence of over-sensitivity to specific background contexts. In contrast, the custom model with a much smaller 49×49 receptive field successfully redetects nearly all copied targets.

The following section discusses our methodology in designing a model better attuned to small low-resolution targets and more resilient to novel background contexts.

## 5. Design of Restricted Receptive Field Architecture

Our approach is designed to address two fundamental challenges in long-range thermal object detection: (1) the *signal challenge* of low resolution, low-PoT target signatures, and (2) the *data challenge* of datasets with limited size and diversity.

Generic state-of-the-art object detection algorithms such as YOLO and Faster R-CNN are not designed with these conditions in mind. The large-scale datasets on which general object detection models are trained and benchmarked, such as ImageNet [58] and COCO [54], are primarily composed of relatively large high-resolution objects compared to the long-range targets in ATR datasets like DSIAC. For general purpose object detection architectures, extracting rich long-distance semantic relationships over the span of an image is of higher priority than avoiding information loss due to early spatial downsampling. However, we conjecture that early downsampling can result in the loss of vital signal information when dealing with very small targets, especially in the infrared. Furthermore, large receptive fields (obtained from early and often spatial downsampling) can exacerbate the problem of overfitting to irrelevant statistical patterns in the data-poor ATR domain.

To address the aforementioned problems, we propose an alternative approach to long-range thermal target detection with two primary design goals:Mitigate information loss due to premature spatial downsampling of the feature maps by reducing the overall degree and frequency of downsampling, thereby extending the processing of fine-grained details.Restrict the network’s receptive field by focusing predictions on a localized area more appropriate to the small size of long-range targets in order to minimize the potential for overfitting.

### 5.1. Spatial Resolution and Receptive Fields

The receptive field of a network layer is the size of the region in the input image, which can inform the activation of a “pixel” in the layer’s output feature map. The receptive field of a convolutional layer’s output feature map is a function of that layer’s kernel size and stride, as well as the receptive field of the previous layer. Using the derivation provided by [59], the receptive field size, $r_{0}$, regarding the input image space for a single-path CNN with *L* layers is
(2)$$r_{0}=\sum_{l=1}^{L}\left(\left(k_{l}-1\right)\prod_{i=1}^{l-1}s_{i}\right)+1$$
where *k* and *s* are a layer’s kernel size and stride, respectively. Downsampling occurs when s>1. The cumulative effect of repeated downsampling results in deep networks with receptive fields that easily cover the entire image.

Skip connections provide a secondary means to increase receptive field size. Feature maps from disparate layers with different spatial resolutions and receptive fields can be combined via upsampling or downsampling. This technique is often used to infuse lower-level localized features with more globalized information from deeper in the network [48,60]. We avoid this strategy in order to enforce a localized context.

Important to note is that Equation (Equation 2) establishes a theoretical upper bound for the size of the region, which can influence an output activation. However, as the receptive field grows, pixels further from the center of a region in the input image have less capacity to influence the output due to having fewer connections with that output, similar to our own foveal vision. The “effective receptive field" of a network is much harder to precisely calculate and depends on activation function, dropout, skip connections, dilation, and input data, among other factors. In general, we can assume the effective receptive field is a Gaussian region occupying only a fraction of the theoretical receptive field [61].

Another important aspect of the architecture’s design is the network stride. Network stride refers to the number of pixels traveled in the original image space for each movement of a convolutional kernel in the final layer. For anchor-based detection methods, the network stride determines the spacing between the anchor boxes. A network with a large stride would have large gaps between the small anchor boxes that have been calibrated for tiny objects. In turn, these gaps place a demand upon the network to produce a larger range of bounding box offset values. This also holds true for some anchor-free methods, which regress offset values in reference to keypoints tiled across the image. Minimizing the number of downsampling layers also minimizes the network’s stride. In this way, the scale of feature extraction and prediction can be aligned with the expected scale of targets.

### 5.2. Receptive Fields for Small Targets

Generic object detection algorithms are commonly benchmarked against the popular COCO dataset [54]. It is also standard practice for models to be pretrained on ImageNet [58] prior to training on COCO. Understandably, general purpose architectures are designed with a bias toward the data distributions of these large-size datasets. In datasets like ImageNet and COCO with medium- and large-size objects, expansive receptive fields help to identify object boundaries. A more holistic view of an image enables better overall scene understanding and, ideally, better predictions.

As objects become smaller, spatial context can provide vital clues for detection. Hu and Ramanan [62] show the importance of contextual information when attempting to detect extremely small faces. Having enough context to recognize the body to which a face belongs is a useful cue, as is noticing the surrounding crowd. In the case of ATR, important contextual information might be a road, tire tracks, or a plume of engine exhaust. How much spatial context is useful depends on the scenario. COCO is composed of approximately 41% small objects (<$32^{2}$ pixels), 34% medium ($32^{2}$ to $96^{2}$ pixels), and 24% large (>$96^{2}$ pixels). The challenging long-range targets in DSIAC are well below the $32^{2}$ upper limit for small objects. From Figure 4, we see that, beyond 3000 m, PoTs are less than $16^{2}$. Therefore, we seek a spatial context sufficient to help in the detection of very small targets, yet limited enough to encourage the network to learn the actual target signature instead of extraneous background features. The bounding box statistics of the data, presented in Table 3, help guide the choice of the receptive field. The smallest square window that can contain the largest target at the closest range (1000 m) is 88×88, while 18×18 encompasses the largest target at the farthest range (5000 m).

For comparison, a list of the receptive fields of other common object detection backbone architectures is provided in Table 4. Either through downsampling or skip connections, most of the baseline models’ theoretical receptive fields cover nearly the entire image. We instead design several custom architectures with much smaller receptive fields more appropriate for small long-range targets.

### 5.3. Receptive Field Ablation

In order to ascertain which receptive field sizes are best for different target ranges and sizes, we create a series of custom one-stage backbone architectures. These custom backbones are similar to the YOLOv3/Darknet family of architectures but differ in the actual number and arrangement of convolutional layers as well as when and how much downsampling is performed. The basic repeated architectural module is a sequence of residual “squeeze and expansion” bottlenecks composed of 3 × 3 and 1 × 1 convolutions, followed by a 3 × 3 convolution with a stride of 2 for downsampling. To isolate the effect of receptive field size on performance, our custom-designed backbones eschew any use of feature pyramid network (FPN) [48] or path aggregation network (PANet) [49] modules, which recombine features from different layers. We investigate receptive fields ranging from 41 × 41 to 965 × 965.

An important model characteristic that varies along with the receptive field size is the model size (the total number of model parameters) and model depth (the number of non-linear transformations). Controlling for model depth is impossible as it is intrinsically linked with the receptive field size. Controlling for model size is possible with careful manipulation of the number of convolutional filters in each layer; however, attempting to do so would still not create a perfectly fair comparison as the number of parameters in each layer would still differ from backbone to backbone. For these reasons, we opt for a traditional progression of layer sizes rather than attempting to control for model size.

For each architecture, we train one model for each of the 5 cross-validation folds and report the average mAP (see Section 4.1, Equation (Equation 1)) and standard deviation. The models are trained on the full set of training data (without any pre-training) for 300 epochs with batch sizes of 32 using the Ultralytics YOLOv5 framework.

Table 5 presents the receptive field ablation results for each testing range. The results clearly indicate our small receptive field architectures (41 and 49 pixels${}^{2}$) are better at detecting very small long-range targets (3.5 and 4.5 km) than the larger receptive field models. For the closer yet challenging 2.5 km scenario, the 105×105 receptive field achieves the best performance. The average width × height of 2.5 km targets is 22.9 × 9.5 pixels, respectively, which is approximately 20% of the $105^{2}$ receptive field. For 3.5 and 4.5 km, the average sizes are 17×6.4 and 12.6×4.9, respectively, or about 35% of the $41^{2}$ receptive field (at the farthest ranges, the actual targets tend to be a few pixels smaller than their ground truth bounding boxes). Given that the effective receptive field is smaller than the theoretical receptive field [61], we observe good results with receptive fields that are very similar in scope to a close-to-medium-size crop around the target. For visual reference, Figure 5 overlays 49×49 and 153×153 windows over a target at varying ranges.

### 5.4. Multi-Head Architecture Ablation

The results of the receptive field ablation presented in Table 5 suggest no single receptive field is sufficient to detect both medium- and long-range targets. Instead, an architecture that performs predictions at multiple scales is better suited to cover the range of target sizes. A multi-headed network architecture allows for predictions at multiple scales of feature resolution. In this section, we present ablation results on a variety of custom multi-headed designs.

We additionally test the incorporation of more sophisticated modules. The cross-stage partial (CSP) module [53] is a dense block [63] that replaces the standard residual block. Densely connected blocks serve a similar function as residual connections, except the input feature map is concatenated to the input of all subsequent layers within the same block of layers, whereas a residual connection is an element-wise addition of the input and output feature maps of the immediate layer. The CSP module is a modification to the dense block that provides more efficient computation of gradients. We also evaluate fusing the deeper, lower resolution feature maps of the 105 receptive field prediction head with the features of the earlier higher-resolution prediction head using an FPN.

Table 6 provides the results of the multi-head ablation study. The best overall performance is achieved by a dual-head architecture with a 105 receptive field prediction head and a 41 receptive field head augmented with upscaled features via an FPN. Unfortunately, no architecture was able to achieve the highest accuracy at every range. Nevertheless, the best overall multi-head model was able to improve performance on the very challenging 4.5 km range by an additional 3% while maintaining detection rates comparable to the best-performing single-head models for each scenario.

### 5.5. Proposed Architecture

Based on the results of our architectural design ablation experiments, we propose a dual-head architecture. The final prediction layer has a receptive field of 105 pixels, the features from which are upscaled and merged with a 41 receptive field layer. A high-level diagram of the model is illustrated in Figure 6 and the layerwise specifications are detailed in Table 7. Omitted from the table are batch normalization and ReLU activations, which occur at every layer. The number of filters in the output heads (indices 11 and 15 in the table) equates to the size of the output prediction vector: |K| target and 1 background class logit plus 4 bounding box offset coordinates for each of the 3 anchors. The upsampling operation is completed by nearest neighbor.

## 6. Comparative Evaluation Results

In this section, we compare the performance of the proposed method against several object detection models on the DSIAC and AI-TOD datasets.

### 6.1. Implementation Details

The Ultralytics YOLOv5 framework is used to train and evaluate our custom-designed dual-head architecture as well as the Conny25 [31], cfCNN [32], and TinyNet [52] models. Due to resource and time constraints, these models have not been pre-trained on the ImageNet or MS-COCO datasets. We implement the cfCNN network as a one-stage anchor-based object detector instead of using its original patch-based classifier design in order to isolate and fairly compare the performance feature extraction backbone. The architecture is kept the same except that the final global average pooling layer is replaced with a convolutional prediction head outputting class probabilities and bounding box coordinates in the same fashion as the other YOLO models. For the implementation details of the YOLO, RetinaNet, and Faster R-CNN models, see Section 4.2. The mAP performance metric is defined in Section 4.1, Equation (Equation 1). Refer to the following results sections for the dataset-specific anchor box size parameters.

### 6.2. DSIAC Results

The results on the DSIAC dataset represent performance on the long-range ground-based MWIR thermal ATR setting. The training and testing protocol, as described in Section 3.1, is designed to test a model’s robustness target locations and ranges not seen in the training set. The models were trained on all images in the training sets of the 1, 2, 3, 4, and 5 km scenarios. The models were evaluated on the validation sets for each range, with the 1.5, 2.5, 3.5, and 4.5 km scenarios constituting the challenging unseen range scenarios.

The (h,w) anchor box sizes for Conny25, cfCNN, and TinyNet are (10,13), (16,30), (33,23), (30,61), (62,45), and (60,120). For our proposed model, the anchor boxes are (10,13), (16,30), (33,23) and (30,61), (62,45), (60,120) for the first and second prediction layers, respectively. For all other models, the arrangement of anchor boxes is identical to the proposed model, except for an additional set of anchors for the third prediction layer: (116, 90), (156, 198), and (373, 326). Given this setup, the calculated best possible recall for each model is at least 99.5%.

As shown in Table 8, our proposed model significantly outperforms all other architectures at every test range. For the long-range 3.5 and 4.5 km scenarios, it scores 84.4% and 27.3% mAP, respectively. On those ranges, the performance of the next best model, TinyNet (originally designed for remote sensing applications), is 82.2% at 3.5 km and 23.7% at 4.5 km. However, TinyNet is less competitive at the 1.5 and 2.5 km scenarios. Critically, our model also demonstrates improved generalizability to the completely novel view of the valley presented in the 2.5 km scenario, reaching 67.2% mAP (a 13% improvement over FRCNN-50’s second highest score). We attribute this success to our architecture’s restricted receptive fields, which focus the network’s attention on the target’s signature.

Our proposed model’s accuracy comes at the cost of speed. This is a consequence of performing predictions on high-resolution feature maps. Our model has a 46 milisecond inference time (using the Nvidia V100). As comparison, TinyNet operates at 1.6 ms, Yolov5 s at 2.5 ms, and FRCNN-101 at 40 ms. As such, our proposed method is not suitable for real-time applications but can operate at speeds similar to the Faster R-CNN two-stage detector.

### 6.3. AI-TOD Results

To evaluate the generalizability of our approach to different scenarios, we utilize the AI-TOD dataset. As discussed in Section 2.4, images in the AI-TOD dataset (all of which are visible-wavelength) have been sourced from several different drone and satellite datasets but selected to only include images with very small targets. In this way, it is an ideal choice for evaluating generalizability to a wide variety of scenarios.

The ‘train’ and ‘val’ sets (comprised of 11,214 and 2167 images, respectively) are used to determine appropriate stopping times for each model. The reported results reflect performance on the ‘test’ set (14,018 images) after training on the ‘trainval’ set (also 14,018 images). The images are all resized to 640 × 640. TinyNet, cfCNN, and Conny25 are trained for 600 epochs. The proposed RF41 + 105 and YOLOv3 models are trained for 650 epochs. The YOLOv5 models are trained for 700 epochs. We report the mAP at both 0.5 IoU and averaged over 0.5 to 0.95 IoU at an interval of 0.5, as well as the precision and recall. Table 9 contains the results.

The proposed model achieves the best performance at 66.39% mAP${}^{\mathrm{IoU}=0.5}$, followed by YOLOv5 x at 64.04% (which performed worst on the DSIAC dataset). The extremely lightweight TinyNet architecture, attaining only 28.07%, is unable to effectively capture the diversity of target appearances and backgrounds in the AI-TOD dataset. Conversely, the much deeper YOLOv5 x is well-suited to large diversified datasets. This was especially evident when examining its increase in performance from the ‘train/val’ set (47.17%). Comparatively, YOLOv5 s only increased from 45.23% to 54.87%, strongly indicating the advantage of a larger-capacity network on larger datasets. Furthermore, the benefits gained by reducing the frequency and immediacy of downsampling on DSIAC’s thermal imagery are less apparent on the visible-wavelength drone and satellite data.

These results validate the effectiveness of the proposed approach on different tiny object detection scenarios, although the trade-off in inference time becomes less justifiable when dealing with more detailed (i.e., visible-wavelength), higher-resolution, and well-varied data. In such situations, a middle ground between our proposed design and a more traditional backbone may be warranted.

## 7. Conclusions

Thermal long-range ATR is a challenging research area due to the nature of the problem as well as the limited data with which to train and evaluate models. The DSIAC dataset is a valuable asset for ATR research, but high correlation between the frames within a video sequence has led to high detection rates in prior works, even at extreme distances when distinguishing the class or even location of the target appears near impossible. We have proposed a challenging protocol for the DSIAC dataset that evaluates model generalizability to video sequences (and thereby target ranges) not available during training, affording a better measure of model performance and robustness. We also observe a phenomenon of background overfitting exacerbated by the very large receptive fields of existing CNN feature extraction backbones. Our proposed architecture, which achieves state-of-the-art performance on the DSIAC dataset, improves detection accuracy and mitigates overfitting by processing the fine details of narrowly focused image regions.

Furthermore, the proposed method is shown to generalize to a variety of different tiny object detection scenarios. It achieves state-of-the-art performance on the tiny object detection AI-TOD dataset composed of drone and satellite visible imagery from a variety of sources. However, our model also sees less improvement over existing methods on the AI-TOD dataset, likely due to the dataset’s numerous, highly varied, and highly detailed visible imagery.

The primary drawback of our method, and an excellent candidate for future work, is inference time. Other avenues of potential future work include the exploration of anchor-free and two-stage models, alternative loss functions and IoU metrics, and multi-scale feature representations. It is a one-stage design that operates at the speed of a two-stage model. Nevertheless, for non-real-time small-thermal-target detection, the proposed RF41+105 FPN model is shown to be highly effective.

## Figures and Tables

**Figure 1 sensors-23-07806-f001:**
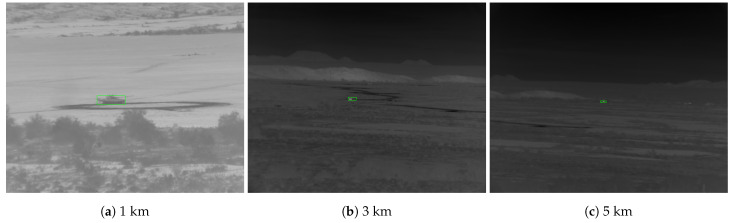
Example images from the Defense Systems Information Analysis Center (DSIAC) dataset of T72 tank during daylight at various ranges and orientations. Ground truth bounding boxes shown in green. The difference in brightness levels is due to min–max normalization.

**Figure 2 sensors-23-07806-f002:**
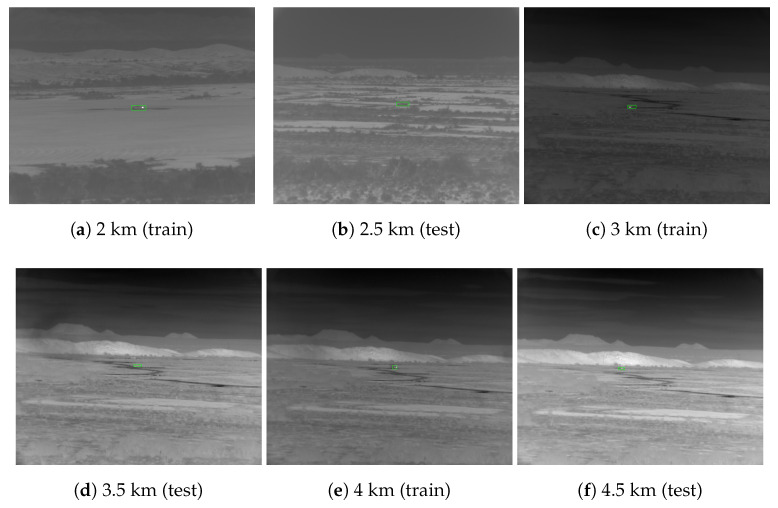
The 3 to 4.5 km scenarios share the same view of the valley, whereas the 2 km, 2.5 km, and 3 km scenarios all have different backgrounds. We hypothesize the drop in performance at the 2.5 km range is likely due to the novel background view, and, consequently, the large receptive fields of the baseline networks invite the memorization of unrelated features.

**Figure 3 sensors-23-07806-f003:**
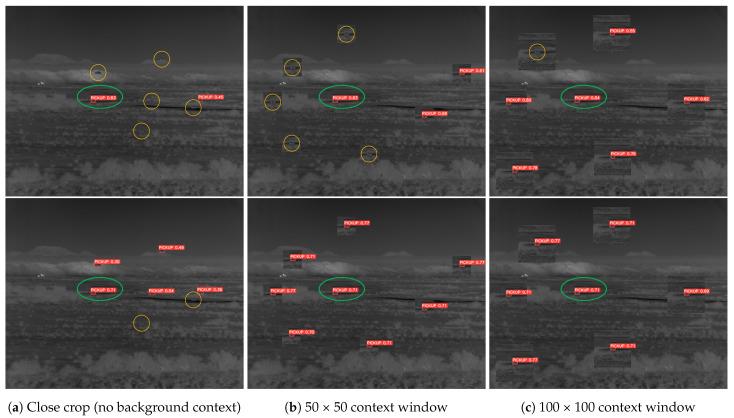
A successfully detected target (green circle) is copy-pasted into other regions of the same image with an increasing amount of background context, from a close crop traced around the target signature (column a) to a 100 × 100 crop around the target (column c). Inference is re-run on the modified images. If a copied target is “redetected”, it is annotated in red; failed redetections are circled in orange. YOLOv5 s (top) has difficulty redetecting the target copies until it appears in a 100 × 100 region of its original background. The 49×49 receptive field model (bottom) successfully redetects nearly all copies, indicating a better ability to learn background-agnostic target representations.

**Figure 4 sensors-23-07806-f004:**
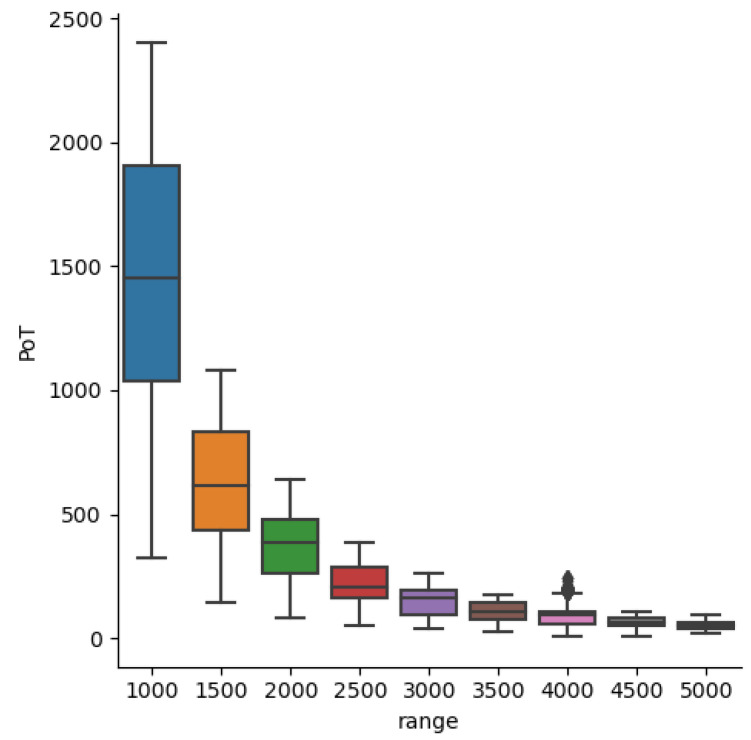
Distribution of pixels on target (PoT) by range for DSIAC’s thermal MWIR imagery.

**Figure 5 sensors-23-07806-f005:**
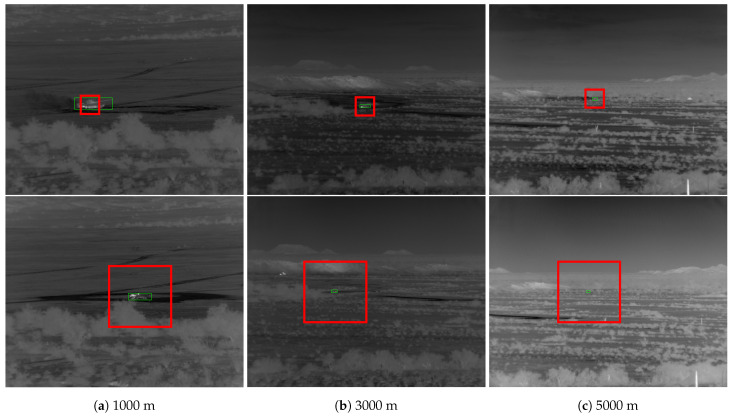
Visualization of 49×49 (top) and 153×153 (bottom) theoretical receptive field (red) overlayed on targets (green) at various distances. DSIAC MWIR imagery is 640×480.

**Figure 6 sensors-23-07806-f006:**
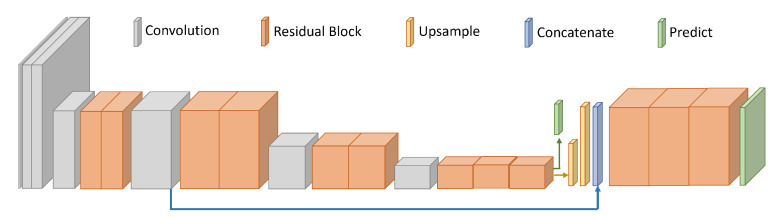
High-level architectural diagram of the proposed RF41+105 FPN network.

**Table 1 sensors-23-07806-t001:** The average number of images per cross-validation fold according to our proposed protocol for the DSIAC dataset. The evaluation set for the training and validation ranges is also referred to as the validation set. Similarly, the evaluation set for the testing ranges is referred to as the test set.

Range	Train	Evaluation
1000	1843	461
1500	-	458
2000	1834	458
2500	-	425
3000	1728	432
3500	-	425
4000	1709	427
4500	-	413
5000	1601	400

**Table 2 sensors-23-07806-t002:** mAP${}^{\mathrm{IoU}=0.5}$ (%) on each test and validation range, averaged over 5 cross-validation folds. Models were trained on training images from the combined set of training ranges (1, 2, 3, 4, and 5 km). Results on the test ranges represent evaluation on image frames from video sequences not used in training. Results on the validation ranges involve training and evaluating on the same set of video sequences (albeit different frames). The results on the validation ranges corroborate the high detection rates reported in prior studies involving DSIAC. However, performance on the unseen test ranges exposes the true difficulty of the task. The highest mAP for each test range is shown in bold.

Model	Test Range				Validation Range				
	**1.5 km**	**2.5 km**	**3.5 km**	**4.5 km**	**1 km**	**2 km**	**3 km**	**4 km**	**5 km**
YOLOv3	97.0	47.6	65.3	15.2	100.0	100.0	99.9	94.5	86.3
YOLOv5 s	96.6	44.8	**77.2**	19.4	100.0	100.0	99.9	93.3	75.7
YOLOv5 m	96.1	40.6	65.9	13.9	100.0	100.0	99.3	92.1	78.1
YOLOv5 l	**98.1**	41.8	50.7	9.1	100.0	100.0	99.9	94.6	86.4
YOLOv5 x	83.8	26.1	27.3	6.8	92.7	88.8	80.1	71.4	57.9
FRCNN-50	88.8	**54.2**	76.8	18.7	100.0	100.0	99.2	92.6	71.9
FRCNN-101	93.1	53.0	66.1	100.5	100.0	100.0	98.9	92.8	77.0
RetinaNet-50	91.0	28.2	60.1	**20.8**	99.7	98.6	93.8	78.2	54.4
RetinaNet-101	80.0	19.1	57.6	10.2	100.0	99.4	94.8	89.8	73.1

**Table 3 sensors-23-07806-t003:** Bounding box pixel size statistics of thermal MWIR DSIAC targets.

Range	Width			Height		
	**Max**	**Min**	**Mean**	**Max**	**Min**	**Mean**
1000	88	18	60.8	30	18	23.7
1500	58	12	39.6	20	12	15.7
2000	44	8	30.3	16	8	11.7
2500	34	6	22.9	12	8	9.5
3000	28	6	19.2	10	6	7.6
3500	24	4	17.0	8	4	6.4
4000	22	2	14.2	8	2	5.5
4500	18	2	12.6	6	2	4.9
5000	18	4	12.2	6	4	4.4

**Table 4 sensors-23-07806-t004:** Theoretical receptive fields of common object detection backbone architectures. RetinaNet [24] and Faster R-CNN [22] both use ResNet backbones.

CNN	Receptive Field
MobileNet	315
ResNet50	483
ResNet101	1027
YOLOv3	965
YOLOv5	725

**Table 5 sensors-23-07806-t005:** mAP${}^{\mathrm{IoU}=0.5}$ (%) of custom YOLOv3-like architectures with varying receptive fields (RF). The best performing architectures for each range are shown in bold.

RF	Test Range			
	**1.5 km**	**2.5 km**	**3.5 km**	**4.5 km**
41	97.3 ± 0.6	52.8 ± 3.0	**86.7** ± **1.4**	**24.5** ± **2.2**
49	98.4 ± 0.6	59.0 ± 2.1	**86.3** ± **0.9**	22.1 ± 2.9
51	98.6 ± 0.5	55.6 ± 2.7	85.0 ± 2.2	20.3 ± 1.3
53	98.4 ± 0.5	52.1 ± 2.2	85.6 ± 1.0	19.7 ± 1.7
73	**99.5** ± **0.4**	63.6 ± 2.6	82.0 ± 1.1	16.5 ± 3.5
105	**99.3** ± **0.4**	**67.4** ± **2.4**	77.3 ± 3.2	14.5 ± 2.7
153	99.0 ± 0.4	66.4 ± 2.4	69.3 ± 2.2	17.6 ± 1.7
213	98.0 ± 0.7	54.2 ± 2.2	53.6 ± 3.9	13.7 ± 3.8
581	91.7 ± 2.5	50.0 ± 2.8	71.5 ± 2.6	18.9 ± 2.8
965	82.9 ± 4.1	39.6 ± 5.3	79.2 ± 2.3	13.2 ± 2.2

**Table 6 sensors-23-07806-t006:** mAP${}^{\mathrm{IoU}=0.5}$ (%) of multiheaded versus single-receptive-field architectures. The head receptive field (RF) specifies the receptive fields of the output prediction heads of the networks. A ★ indicates the native receptive field of the prediction head has been augmented by upscaled features with larger receptive fields via the feature pyramid network (FPN). CSP denotes the use of cross-stage partial dense blocks in place of residual bottleneck blocks. The overall average is computed as the mAP for each individual test range divided by the number of test ranges.

Head RF				FPN	CSP	Test Range				Overall
**41**	**49**	**73**	**105**			**1.5 km**	**2.5 km**	**3.5 km**	**4.5 km**	**Avg.**
•				-	-	97.3 ± 0.6	52.8 ± 3.0	**86.7** ± **1.4**	24.5 ± 2.2	65.3
	•			-	-	98.4 ± 0.6	59.0 ± 2.1	**86.3** ± **0.9**	22.1 ± 2.9	66.5
		•		-	-	**99.5** ± **0.4**	63.6 ± 2.6	82.0 ± 1.1	16.5 ± 3.5	65.4
			•	-	-	99.3 ± 0.4	67.4 ± 2.4	77.3 ± 3.2	14.5 ± 2.7	64.6
•			•			98.8 ± 0.6	60.6 ± 2.3	86.1 ± 1.2	26.5 ± 1.2	68.0
★			•	✓		**99.5** ± **0.3**	67.2 ± 2.2	84.4 ± 1.2	**27.3** ± **2.9**	**69.6**
•			•		✓	98.8 ± 0.4	62.4 ± 3.2	84.9 ± 0.6	26.6 ± 2.1	68.2
★			•	✓	✓	99.3 ± 0.5	65.3 ± 5.2	82.5 ± 0.8	**27.1** ± **1.7**	68.6
	•		•			99.0 ± 0.5	64.0 ± 1.2	**86.3** ± **1.2**	24.0 ± 1.4	68.3
	★		•	✓		**99.6** ± **0.2**	64.6 ± 3.6	85.3 ± 1.1	**27.4** ± **2.0**	**69.2**
	•		•		✓	98.6 ± 0.5	61.6 ± 2.9	84.7 ± 2.1	23.6 ± 2.5	67.1
	★		•	✓	✓	99.4 ± 0.2	64.3 ± 2.9	85.6 ± 1.4	24.4 ± 2.5	68.4
•		•	•			**99.5** ± **0.3**	68.2 ± 3.3	85.6 ± 1.5	21.3 ± 3.0	68.7
★		★	•	✓		99.0 ± 0.2	**71.3** ± **2.7**	80.9 ± 2.1	16.8 ± 1.7	67.0
•		•	•		✓	98.9 ± 0.6	66.7 ± 2.1	82.3 ± 1.4	18.5 ± 1.8	66.6
★		★	•	✓	✓	98.8 ± 0.2	68.0 ± 2.4	79.7 ± 2.4	17.9 ± 2.4	66.1

**Table 7 sensors-23-07806-t007:** Layerwise specifications for RF 41 + 105 FPN architecture. *K* is the set of target classes. Layer indices 4, 6, 8, 10, and 14 indicate repeated blocks of (Conv, Conv, Residual) layers. Output heads are shown in bold.

Idx		Type	Filters	Size	RF
1		Conv	32	3 × 3	1
2	2×	Conv	64	3 × 3	5
3		Conv	128	3 × 3/2	7
		Conv	64	1 × 1	
		Conv	128	3 × 3	
4	2×	Residual			17
5		Conv	256	3 × 3	21
		Conv	128	1 × 1	
		Conv	256	3 × 3	
6	2×	Residual			29
7		Conv	256	3 × 3/2	33
		Conv	128	1 × 1	
		Conv	256	3 × 3	
8	2×	Residual			49
9		Conv	256	3 × 3/2	57
		Conv	128	1 × 1	
		Conv	256	3 × 3	
10	3×	Residual			105
11		ConvHead	3 × (|K| + 5)	1 × 1	105
12		Upsample(11)			
13	2×	Concat(6,12)			
		Conv	128	1 × 1	
		Conv	256	3 × 3	
14	3×	Residual			41 *
15		ConvHead	3 × (|K| + 5)	1 × 1	41 *

* denotes receptive field is augmented by upscaled features.

**Table 8 sensors-23-07806-t008:** mAP${}^{\mathrm{IoU}=0.5}$ (%) on each test and validation range, averaged over 5 cross-validation folds. The highest mAP for each test range is shown in bold.

Model	Test Range				Validation Range				
	**1.5 km**	**2.5 km**	**3.5 km**	**4.5 km**	**1 km**	**2 km**	**3 km**	**4 km**	**5 km**
YOLOv3	97.0	47.6	65.3	15.2	100.0	100.0	99.9	94.5	86.3
YOLOv5 s	96.6	44.8	77.2	19.4	100.0	100.0	99.9	93.3	75.7
YOLOv5 m	96.1	40.6	65.9	13.9	100.0	100.0	99.3	92.1	78.1
YOLOv5 l	98.1	41.8	50.7	9.1	100.0	100.0	99.9	94.6	86.4
YOLOv5 x	83.8	26.1	27.3	6.8	92.7	88.8	80.1	71.4	57.9
FRCNN-50	88.8	54.2	76.8	18.0	100.0	100.0	99.2	92.6	71.9
FRCNN-101	93.1	53.0	66.1	10.5	100.0	100.0	98.9	92.8	77.0
RetinaNet-50	91.0	28.2	60.1	20.0	99.7	98.6	93.8	78.2	54.4
RetinaNet-101	80.0	19.1	57.6	10.2	100.0	99.4	94.8	89.8	73.1
Conny25 [31]	96.3	38.9	54.6	10.0	100.0	99.9	98.9	88.6	62.7
cfCNN [32]	96.1	44.8	66.4	14.1	99.9	99.9	99.4	90.0	62.3
TinyNet [52]	89.1	48.0	82.2	23.7	100.0	100.0	99.7	91.3	68.4
**RF41+105 FPN (Ours)**	**99.5**	**67.2**	**84.4**	**27.3**	100.0	100.0	99.7	95.2	87.4

**Table 9 sensors-23-07806-t009:** mAP, precision (P), and recall (R) on the test set of the AI-TOD dataset. Bolded values indicate the best performance in their respective categories.

Model	mAP${}^{0.5}$	mAP${}^{0.5:0.95}$	P	R
YOLOv3	61.82	30.6	77.22	59.9
YOLOv5 s	54.87	23.22	69.72	54.87
YOLOv5 m	60.64	27.9	73.59	59.47
YOLOv5 l	62.49	31.22	**77.25**	61.25
YOLOv5 x	64.04	32.55	76.83	62.86
Conny25 [31]	23.74	8.12	51.37	25.05
cfCNN [32]	20.5	7.05	73.73	20.32
TinyNet [52]	28.07	10.21	40.7	30.99
RF41 + 105 FPN	**66.39**	**32.94**	69.12	**67.95**

## Data Availability

Publicly available datasets were analyzed in this study. The DSIAC dataset is available at: https://dsiac.org/databases/atr-algorithm-development-image-database/. The AI-TOD dataset is available at: https://github.com/jwwangchn/AI-TOD. The training and testing protocol for the DSIAC dataset is provided in the Appendix A.

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
