# Peer review of "Long-Range Thermal Target Detection in Data-Limited Settings Using Restricted Receptive Fields"

_sensors, 2023, doi:10.3390/s23187806_

Round 1
Reviewer 1 Report
This paper conducted a thorough study on small target detection of long-range thermal imagery based on the deep learning method. The proposed feature extraction architecture aimed to address the issue of the limited size and variability of the available datasets. The result was quantitatively compared with generic baseline models. The paper is well-organized and written and minor corrections are expected. Please consider addressing the following comments:
In section 3.2, the paper mentions the pixel intensity is converted into the 8-bit format. Is any other preprocessing method used to handle outlier data (such as abnormal temperature points) in the thermal imagery?
In line 317, it recommends adding figures to illustrate the “two common points of failure” so that the reader could have a better understanding of the task difficulty when using general DL architecture.
Author Response
We are incredibly appreciative of the time and effort given by the reviewer to read and provide comments on our work. We are especially grateful for the encouraging words and thoughtful feedback that was given. The following is a summary of the reviewer comments (in bold) and our attempts at addressing each of them:
In section 3.2, the paper mentions the pixel intensity is converted into the 8-bit format. Is any other preprocessing method used to handle outlier data (such as abnormal temperature points) in the thermal imagery?
No other preprocessing methods were used to handle outlier data. This has now been made explicit in the text. (Line 341-342)
In line 317, recommend adding figures to illustrate the “two common points of failure” so that the reader could have a better understanding of the task difficulty when using general DL architecture.
We attempted to include another figure to illustrate the two points of failure, however this caused too much distance between the tables/figures and the text which discussed them. Instead we point to the two sections in the text which discuss each point of failure in more detail. The first section references the Table 2 which shows the very low mAP scores in the 4.5km scenario. The second section references the problem of background overfitting, which includes an illustrative figure.
Reviewer 2 Report
Review report:
• A brief summary (one short paragraph) outlining the aim of the paper, its main contributions and strengths
Paper presents address the problem of long-range target detection in thermal infrared imagery that is a challenging problem due to the low resolution and limited detail captured by thermal sensors. Authors proposed a novel convolutional neural network (CNN) feature extraction architecture designed for small object detection in data-limited settings with focus on long-range, ground-based, thermal vehicle detection, also with effectiveness on drone and satellite aerial imagery. Authors proposed a design of a challenging train/test protocol for the DSIAC dataset. Performance analysis of proposed model was made alongside state of the art object detection models on the DSIAC and Tiny Object Detection in Aerial Images (AI-TOD) datasets. Proposed approach achieved state-of-the-art results on the 2 datasets - the Defense Systems Information Analysis Center (DSIAC) automated target recognition (ATR) and the Tiny Object Detection in Aerial Images (AI-TOD) datasets.
• General concept comments
Main observations are:
- bibliography contains 60 references, 8 older than 10 years, 24 references from last 10 years and 26 from last 5 years
- bibliography has 2 references with missing year (10, 51)
- bibliography should be improved with newer references
- English language could be improved
- there are some misspelling in the text
- one equation is not referenced in text
• Specific comments referring to line numbers, tables or figures that point out inaccuracies within the text or sentences that are unclear.
- in Keywords, line 15 "thermal infrared.)" should be "thermal infrared."
- in 1. Introduction, line 59, "SOTAalgorithms" should be "SOTA algorithms"
in 4.1. Performance metric, line 326, equation 1 is not referenced in text
• Is the manuscript clear, relevant for the field and presented in a well-structured manner?
Manuscript is relevant for the field and well presented.
• Are the cited references mostly recent publications (within the last 5 years) and relevant? Does it include an excessive number of self-citations?
Cited references are relatively new, out of 60 references, 8 are older than 10 years, 24 references are from last 10 years and 26 are from last 5 years.
Bibliography has 2 references with missing year (10, 51).
Bibliography should be improved with newer references.
• Is the manuscript scientifically sound and is the experimental design appropriate to test the hypothesis?
The experiment design is appropriate to test the hypothesis.
• Are the manuscript’s results reproducible based on the details given in the methods section?
More information should be added in order for the results to be reproducible.
English language can be improved.
Author Response
We are incredibly appreciative of the time and effort given by the reviewer to read and provide comments on our work. We are especially grateful for the encouraging words and thoughtful feedback that was given. The following is a summary of the reviewer’s comments (in bold) and our attempts at addressing each of them.
- in Keywords, line 15 "thermal infrared.)" should be "thermal infrared."
This has been corrected.
- in 1. Introduction, line 59, "SOTAalgorithms" should be "SOTA algorithms"
This has been corrected.
- in 4.1. Performance metric, line 326, equation 1 is not referenced in text
Equation 1 is now referenced in the following sections: 4.3, 5.3, and 6.1
Bibliography has 2 references with missing year (10, 51).
The dates which the resources in question were accessed has been added.
Bibliography should be improved with newer references.
There is limited research being done specifically on long-range thermal ATR. Furthermore, some of the recent ATR papers we found we thought were lacking in quality and thus were excluded. However, we did add a discussion of three papers all published within the last year: two ATR papers using the DSIAC dataset, and one paper published in Nature this July on thermal detection and ranging (Lines 187-196). If there are any specific relevant works you feel we have overlooked, we would be happy to incorporate them into the paper.
More information should be added in order for the results to be reproducible.
We have included the exact training and testing sets in the form of supplementary files containing the image filenames. This is now mentioned in the dataset description (Line 314-315).
Reviewer 3 Report
Detection of long-range targets based on infrared infrared images is an important scientific and research problem. Due to the relatively low resolution of thermal images, the amount and quality of the information obtained is limited. This is due to the specificity of thermovision, characterized by, among others, small amount of detail recorded by thermal sensors. In order to identify thermal information contained in digital images, artificial intelligence methods are increasingly used, in particular neural image analysis techniques. In recent years, modern techniques of deep learning of convolutional neural networks (CNN) have been used for this purpose.
The authors proposed the evolution of the dynamically developed information technology based on convolutional neural networks. They dedicated the method as a tool for quick detection of small civilian and military targets, presented in the form of digital thermal imaging images. The authors solved machine learning problems in the context of generating adequate CNN topologies, resulting mainly from the specificity of thermal images (number and quality of images, limited information density, tendency to learn the network "by heart", etc.).
The proposed approach is characterized by a significant increase in efficiency in relation to the methods used so far. It can therefore be dedicated as support for activities carried out, e.g. in the automated Center for the Analysis of Information on Defense Systems (DSIAC), it can support the process of target recognition (ATR) and generate data sets for detecting small objects in aerial photographs (AI-TOD). The authors also revealed the disadvantage of the presented identification technique in the form of a long time of generating results by a convolutional neural model and indicated the directions of work aimed at further improvement of the proposed method.
The work is methodologically correct and contributes to the development of scientific knowledge. It also has a strongly accentuated utilitarian thread. My minor remarks concern the suggested stylistic and punctuation corrections:
- complete the descriptions of the quantities used in the formulas: (1)(2)
- describe in more detail the structure of training sets (training, test and validation)
- despite the fact that the neural models used are widely known, it is worth adding an elementary description of the topology of the CNNs used (maybe in graphic form).
- fig.3. - description a)b)c) is inadequate to the whole drawing: I propose to correct the description of the drawing.
Author Response
We are incredibly appreciative of the time and effort given by the reviewer to read and provide comments on our work. We are especially grateful for the encouraging words and thoughtful feedback that was given. The following is a summary of the reviewer’s comments (in bold) and our attempts at addressing each of them.
- describe in more detail the structure of training sets (training, test and validation)
We have included the exact training and testing sets in the form of supplementary files containing the image filenames. (Line 314-315)
- despite the fact that the neural models used are widely known, it is worth adding an elementary description of the topology of the CNNs used (maybe in graphic form).
We added brief descriptions of the CNN topologies (Line 328-339). Adding more figures is difficult due to this creating too large of a distance between the remaining figures/tables and the sections which describe them.
- complete the descriptions of the quantities used in the formulas: (1)(2)
We apologize but we are uncertain as to which quantities need descriptions? If you can please explain what exactly is unclear or missing, we will fix the issue.
- fig.3. - description a)b)c) is inadequate to the whole drawing: I propose to correct the description of the drawing.
We've attempted to improve the description of the figure.
Round 2
Reviewer 2 Report
Thank you very much for your effort, all observations were addressed and all recommendations have been implemented.
English language was improved in the new version.